# Debunking the Illusion of Backdoor Robustness in Vision Transformers

## Abstract

Backdoor attacks, which make Convolution Neural Networks (CNNs) exhibit specific behaviors in the presence of a predefined trigger, bring risks to the usage of CNNs. These threats should be also considered on Vision Transformers. However, previous studies found that the existing backdoor attacks are powerful enough in ViTs to bypass common backdoor defenses, *i.e.*, these defenses either fail to reduce the attack success rate or cause a significant accuracy drop. In this paper, we first investigate this phenomenon and find that this kind of achievement is over-optimistic, caused by the inappropriate adaptation of defenses from CNNs to ViTs. Existing backdoor attacks can still be easily defended against with proper inheritance. Furthermore, we propose a more reliable attack: adding a small perturbation on the trigger is enough to help existing attacks more persistent against various defenses. We hope our contributions, including the finding that existing attacks are still easy to defend with adaptations and the new backdoor attack, will promote more in-depth research into the backdoor robustness of ViTs.

## 1 Introduction

Vision Transformers (ViTs) (Dosovitskiy et al., 2021; Liu et al., 2021) have demonstrated outstanding performance in various tasks, including image classification (Yuan et al., 2021; Touvron et al., 2022), semantic segmentation (Strudel et al., 2021), and image generation (Hirose et al., 2021; Bao et al., 2022), leading to their widespread popularity. However, strong performance alone is insufficient for ViT to be practically deployable. It must also exhibit trustworthiness without posing severe security risks, and one of the most notable ones is the backdoor attacks (Gu et al., 2017; Chen et al., 2017), which implant unexpected behaviors inside models,

Table 1: The performance of FT against Badnets attack for ResNet-18 and ViT-B on CIFAR-10 (Wu et al., 2022).

|     | ResNet18 | ViT-B  |
| --- | -------- | ------ |
| ASR | 1.48%    | 8.81%  |
| ACC | 89.96%   | 42.00% |

making the victim model produce specific misclassification in the presence of a predefined trigger while maintaining high performance on benign images. While previous studies mainly focus on Convolutional Neural Networks (CNNs), there is a growing need for an in-depth investigation of ViTs to help practitioners better understand the potential risks and deploy them more reliably.

After a long arms race between backdoor attack and defense, for CNNs, a relatively simple defense has the potential to make backdoor attacks fail. Taking the fine-tuning defense and Badnets attack in Table 1 as an example, we find that after performing the defense, the attack success rate (ASR) on ResNet18 only has 1.48% while the benign accuracy (ACC) is 89.96%, indicating a failure of the attack. Contrastingly, ViTs, when subjected to the same attack, display an increased ASR and decreased ACC, implying the disruption of the benign utility. Given that Badnets is model-agnostic, this differential outcome motivates us to explore which factor leads to this underlying disparity between CNNs and ViTs.

Drawing inspiration from Mo et al. (2022), we distinguish a crucial observation: 1) CNNs are usually trained by SGD and its fine-tuning defense is also trained by SGD; 2) ViTs are typically trained by AdamW while its fine-tuning defense is trained by SGD (NOT AdamW, inheriting from earliest work (Dosovitskiy et al., 2021), which first introduces transformers to computer vision). This discrepancy in optimizers raises the possibility that the perceived vulnerability of ViTs (with defense) might be overstated. In this paper, we first conduct a series of experiments to comprehensively investigate the

above hypothesis, which is further confirmed that the threat posed to ViTs with defense has been magnified. Upon minor modifications, ViTs with existing backdoor defense methods demonstrate clear resistance to attacks, mirroring the robustness of CNNs.

To this end, we are wondering whether a more powerful attack exists that can better evade current defenses. Therefore, we analyze backdoored models and further propose a simple yet effective attack. We discover that it is easy for backdoor defenses to detect and utilize the differences in channel activations due to the noticeable difference in the intermediate layers between the inputs with and without triggers. However, we can reduce this difference by adding small perturbations to the triggers before training while keeping triggers unchanged during inference, resulting in more reliable backdoor attacks. Additionally, our method has transferability across different transformer architectures and is effective for both small and large datasets.

In summary, our contributions are summarized as follows:

- We investigate the existing backdoor defenses on ViTs and find the outstanding performance of the backdoor attacks to ViTs is over-estimated due to the inappropriate adaptations. Further, we provide a practical training recipe to improve the performance of current defense and show that existing attacks on ViTs can still be easily resisted.
- We propose to add small perturbations to the triggers before training to suppress the difference in the intermediate-level representations between the inputs with and without triggers, resulting in a more powerful attack, which is termed the channel activation attack in ViT (CAT).
- Our contributions, including the finding of existing attacks to current defenses and the development of a new attack, contribute to a reliable baseline for the backdoor robustness of ViTs. We hope it can be a cornerstone of future studies in improving the backdoor robustness of ViTs.

## 2 RELATED WORK

### 2.1 BACKDOOR ATTACK

Backdoor attacks (Gu et al., 2017; Chen et al., 2017; Souri et al., 2022; Xia et al., 2024), also known as Trojan attacks, indicate the behaviors of implanting specific malicious behavior into machine learning models, which make the models perform well on benign data while leading to specific misclassifications on inputs containing triggers (*i.e.*, triggered inputs). The adversary usually poisons the training data (Zeng et al., 2021) or controls the training process (Liu et al., 2018b) to achieve this. Typically, a trigger pattern is added to the input image as follows,

$$\mathbf{x}_p = (\mathbf{1} - \mathbf{m}) \odot \mathbf{x} + \mathbf{m} \odot \mathbf{t}, \tag{1}$$

where $\mathbf{t}$ is the trigger pattern and mask $\mathbf{m}$ indicates the pixels affected by the trigger pattern. Usually, the adversary re-labels the triggered input as the predefined target class (*i.e.* in a dirty-label setting). Models trained on a mixture of these poisoned data and other benign data are implanted with an unexpected correlation between the trigger pattern and the target class. To improve the attack stealthiness, some studies explored less noticeable trigger designs like the semi-transparent ones (Chen et al., 2017), the elastic transformed ones (Nguyen & Tran, 2021), and the input-aware ones (Nguyen & Tran, 2020). Besides, since incorrect annotation might expose the existence of the poisoned data, some studies focus on attacks without re-labeling (clean-label settings) (Turner et al., 2019; Barni et al., 2019; Shafahi et al., 2018; Zeng et al., 2023; Yu et al., 2024). Although most previous backdoor attacks focus on CNNs, researchers have started to focus on backdoor attacks on ViT regarding their popularity. Although ViTs are reported to be more robust against adversarial attacks (Aldahdooh et al., 2021; Shao et al., 2021) and common corruption (Bai et al., 2021; Bhojanapalli et al., 2021), they are still vulnerable to backdoor attacks (Lv et al., 2021; Subramanya et al., 2022). Reliable attacks are needed to help practitioners properly understand the risks of backdoor threats and deploy these models reliably.

### 2.2 BACKDOOR DEFENSE

To mitigate the potential risks caused by backdoor attacks, numerous studies proposed various defense methods, mainly categorized into **defense during training** and **defense after training**. Defense

during training attempts to mitigate the impact of poisoned data in the training set. Some methods detect and remove poisoned data by treating them as outliers (Chou et al., 2018; Udeshi et al., 2022; Gao et al., 2019), some employ semi-supervised learning to bypass the incorrect correlations (Huang et al., 2022), and others utilize differential privacy to ensure that a poisoned portion of training data is unable to cause severe results (Miao et al., 2022). Meanwhile, the defense after training (Zheng et al., 2022b) includes those that detect the backdoor samples from the model inputs (Tran et al., 2018; Gao et al., 2019) or purify the model to removes the backdoor behavior inside DNNs. Since the latter category is closer to the ideal goal, it has received more attention. This can be accomplished by fine-tuning the model using a small amount of clean data (Sha et al., 2022) (FT) and further improving it with neuron pruning (Liu et al., 2018a) or attention alignment (Li et al., 2021a). Since the performances of fine-tuning are easy to suffer a substantial decrease when the data is limited, another popular method is selectively removing neurons related to the backdoor behaviors (Wu & Wang, 2021; Chai & Chen, 2022; Wang et al., 2019): Built upon the observation that the backdoor behavior can be revealed by the adversarial neuron perturbation, ANP (Wu & Wang, 2021) formulates the following min-max problem with dataset $D_v$ to expose the malicious neuron:

$$\min_{\mathbf{m} \in [0,1]^n} \big[ \alpha \mathcal{L}_{D_v}(\mathbf{m} \odot \mathbf{w}, \mathbf{b})$$
$$+ (1-\alpha) \max_{\delta, \xi \in [-\epsilon, \epsilon]^n} \mathcal{L}_{D_v}((\mathbf{m}+\delta) \odot \mathbf{w}, (1+\xi)\mathbf{b}) \big], \tag{2}$$

where $\delta$ and $\xi$ are the perturbations to the weight $\mathbf{w}$ and bias $\mathbf{b}$ of all neurons respectively. They maximize the cross-entropy loss $\mathcal{L}_{D_v}$ and $\mathbf{m}$ is the mask that adversarially preserves the clean accuracy and covers up the backdoor behavior. Then the neurons corresponding to low mask values are pruned to purify the backdoor model. As an improved approach based on ANP, AWM in (Chai & Chen, 2022) proposes to adopt the element-wise weight and perturb the input data instead to gain better performances on small networks. This paper primarily focuses on defense after training. Because ViTs demand a large amount of data and extensive training resources, it has become impractical for most practitioners to train ViTs from scratch, making defense after training a more realistic scenario. Previous studies (Wu et al., 2022; Yuan et al., 2023) suggested that directly applying defenses from CNNs to ViTs fails. For example, fine-tuning decreases natural accuracy from 94.58% to 42.00% against the Badnets attack and fine-pruning totally collapses in (Yuan et al., 2023). At the meantime, only a few defense methods specially designed for ViT are proposed (Doan et al., 2022; Subramanya et al., 2024) and their performance is lagging far behind the state-of-the-art defense on CNNs: The adaptive defense proposed in (Zheng et al., 2022a) only decreases the ASR of TrojViT (a ViT-specific attack) to 77.13% and the patch processing method in (Doan et al., 2022) fails to detect 33.2% backdoor examples on CIFAR-10. It seems that existing attacks can already obtain outstanding performances on resisting defense for ViTs. However, in this paper, after re-investigating various backdoor defenses with ViTs, we reveal that the achievement obtained by previous attacks is attributed to the improper adaptation of defenses. Furthermore, we provide a new attack, based on the empirical observation of the channel activations to help existing attacks evade defenses more effectively.

## 3 THE VULNERABILITY OF DEFENSE ON ViTs TO EXISTING ATTACKS

In this section, we reevaluate the perceived susceptibility of ViTs to prevailing backdoor attacks when equipped with potential defenses. We primarily consider two categories of defenses: one is fine-tuning-based, including Fine-Tuning (FT), Fine-Pruning (FP) (Liu et al., 2018a), and Neural Attention Distillation (NAD) (Li et al., 2021a), Fine-tuning with SAM optimizer (FT-SAM) Zhu et al. (2023), Super Fine-tuning (Super-FT) Sha et al. (2022) and the other is pruning-based, including Adversarial Neuron Pruning (ANP) (Wu & Wang, 2021) and Adversarial Weight Masking (AWM) (Chai & Chen, 2022).

### 3.1 THREAT MODEL AND BASIC SETTINGS

**Threat Model:** Consistent with the assumptions made by most attacks (Wu et al., 2022), our threat model limits the capability of attackers to the access of training data. An unaware third party trains a ViT model using the poisoned data and helps downstream users who lack the training resources to solve the downstream tasks. The defenders provides service for the users and ensure that the downloaded parameters from the third party are free of backdoor threats.

Table 2: The comparison between SGD and AdamW optimizer on FT. Here, AvgDrop represents the average drop of six attacks on ASR/ACC after performing FT.

| Attack | ACC | | | ASR | | |
|---|---|---|---|---|---|---|
| | No defense | SGD | AdamW | No defense | SGD | AdamW |
| Badnets | 97.85 | 14.32 | 95.14 | 100.00 | 5.02 | 0.72 |
| Blend | 97.85 | 10.99 | 95.32 | 100.00 | 7.10 | 5.22 |
| CLB | 97.83 | 15.67 | 95.15 | 96.23 | 6.80 | 3.39 |
| SIG | 97.50 | 12.62 | 95.32 | 90.57 | 12.12 | 5.66 |
| IAD | 97.79 | 15.00 | 95.57 | 100.00 | 13.80 | 5.36 |
| SSBA | 98.19 | 11.78 | 96.05 | 99.23 | 9.16 | 0.50 |
| AvgDrop | - | 84.44 | **2.41** | - | 88.67 | **94.20** |

**Settings:** We train a backdoored ViT-B (Dosovitskiy et al., 2021) with various attack methods. Specifically, we initialize the model with a pre-trained weight (Wightman, 2019) on the ImageNet-1k (Deng et al., 2009) and then fine-tune it on CIFAR-10[1] (Krizhevsky et al., 2009). Note that a portion of CIFAR-10 training data is poisoned to implant the backdoor behavior, *i.e.*, some images are added with the trigger pattern and are re-labeled as the target class if expected. We apply six commonly-used attack methods: 1) Badnets (Gu et al., 2019), 2) Blend (Chen et al., 2017), 3) CLB (Turner et al., 2019), 4) SIG (Barni et al., 2019), 5) IAD (Nguyen & Tran, 2020), 6) SSBA (Li et al., 2021b). Their trigger design and poisoning method in the original paper are kept. For dirty-label attacks, we set the poison rate as 5% and for clean-label attacks, we poisoned 80% images of the target class. To accommodate the input size of ViT, we first add triggers to CIFAR-10 images ($32 \times 32$) and then resize them to a larger size ($224 \times 224$). For more detailed information, please refer to Appendix B. Here, we use accuracy (ACC) to indicate the classification performance on benign data, and attack success rate (ASR), the percentage of triggered input being classified as the target class, to indicate the attack performance. Note that we will remove the inputs whose ground-truth label is the target class, and thus, a successful defense should make ASR as low as 0.

### 3.2 VITS WITH FINE-TUNING-BASED DEFENSE

Fine-tuning is one of the most basic and model-agnostic defenses. However, as discussed in Section 1, directly inheriting fine-tuning-based defense strategies from CNNs can potentially lead to suboptimal outcomes. Note that SGD is the commonly used optimizer for both training and fine-tuning for CNNs, while for ViTs, the first work (Dosovitskiy et al., 2021) introducing Transformers to computer vision, adopts AdamW for pre-training and SGD for fine-tuning. Notably, prior work (Wu et al., 2022) on backdoor defense naturally inherit this strategy and observes notably diminished accuracy across multiple backdoor attacks. This discrepancy in optimizers motivates us to study the potential influence of optimizers on backdoor defense. The initial learning rates for SGD and AdamW are set to 0.02 and 3e-4, respectively. For the other parameters in AdamW, we use the common settings of the original ViTs (refer to Appendix C for details). Table 2 illustrates the experimental fine-tuning (FT) results against various backdoor attacks. For the results on other fine-tuning-based attacks, please refer to Appendix D. We find that SGD exhibited significant instability on ViTs: Its ACC is below 20% under all attacks. In contrast, AdamW can achieves not only high ACC and but also low ASR under all settings. Therefore, simply using SGD for backdoor defense on ViTs will yield highly unstable performance. In Appendix E, we analyze the reason behind it from an empirical view.

### 3.3 VITS WITH PRUNING-BASED DEFENSE

Pruning is also a typical defense approach, which attempts to remove backdoor-related neurons/channels and is severely impacted by the architectures. In previous studies, pruning-based methods have achieved excellent robustness against backdoor attacks with CNNs (Wu & Wang, 2021; Chai & Chen, 2022). However, when we directly apply these methods to ViTs, we find that they are unable to effectively defend as shown in Table 3. Specifically, ANP fails to reduce ASR and cannot remove the backdoor-related neurons. Besides, although AWM reduces ASR, it also largely decreases ACC, making the model unusable. To explore the potential reason, we look deeply at the implementation of ANP and find that ANP actually prunes channels inside norm layers rather than neurons inside convolutional layers. This is because, in CNNs, each neuron is typically surrounded

---

[1]Ony 95% of the original training data on CIFAR-10 are used to train the backdoored model, and the remaining data are kept for defense.

Table 3: The Performance of Pruning-based Defense with or without adaptation. Here, AvgDrop represents the average drop of six attacks on ASR/ACC after performing the defense.

| Attack | No defense | | ANP | | ANP Adapted | | AWM | | AWM Adapted | |
|---|---|---|---|---|---|---|---|---|---|---|
| | ACC | ASR | ACC | ASR | ACC | ASR | ACC | ASR | ACC | ASR |
| Badnets | 97.85 | 100.00 | 97.85 | 100.00 | 94.26 | 1.34 | 85.98 | 1.24 | 95.02 | 0.71 |
| Blend | 97.85 | 100.00 | 97.85 | 100.00 | 92.70 | 23.70 | 83.29 | 2.03 | 95.08 | 1.70 |
| CLB | 97.83 | 96.23 | 97.83 | 96.23 | 95.71 | 12.71 | 85.67 | 3.48 | 95.60 | 1.52 |
| SIG | 97.50 | 90.57 | 97.50 | 90.57 | 92.60 | 1.48 | 87.22 | 1.16 | 94.58 | 3.87 |
| SSBA | 98.19 | 99.23 | 98.19 | 99.23 | 93.88 | 0.23 | 82.60 | 1.53 | 95.64 | 1.24 |
| IAD | 97.79 | 100.00 | 97.79 | 100.00 | 92.91 | 5.64 | 88.07 | 6.49 | 94.47 | 6.38 |
| AvgDrop | - | - | **0.00** | 0.00 | 4.19 | **90.16** | 12.36 | **95.17** | **2.77** | 95.10 |

(a) Before defense      (b) After FT      (c) After AWM

Figure 1: The average activations for different channels before (a) and after the backdoor defense (b)-(c). The activations are sorted in descending order of the activations on natural samples.

by at least one norm layer[2]. However, in ViT, many norm layers are removed, and norm-layer-based pruning only influences part of neurons and limits the defense performance. Meanwhile, AWM utilizes element-wise masks for optimization, whose number of parameters is the same as the total number of parameters of ViT. Since ViTs are typically larger and lack inductive bias, AWM encounters the severe overfitting issue, leading to low accuracy. Therefore, to make pruning methods applicable to ViTs, selecting appropriate granularity and pruning locations is necessary. Here, we recommend directly pruning all channels of linear projection inside both attention and MLP layers, which provides better coverage than ANP and requires fewer parameters compared to AWM. As shown in Table 3, this modification decreases ASR notably and keeps ACC high.

## 4 PROPOSED BACKDOOR ATTACKS

Following the above analysis, existing defense methods (ViTs adapted) successfully defend against backdoor attacks in ViTs, just as they do in CNNs. Here, we want to explore whether there exist new backdoor attacks to beat the newly adapted defense on ViTs.

To obtain a better insight into why defense methods can detect and remove backdoor behaviors, we investigate the per-channel activations before the MLP head in ViT. We illustrate the average activations of all channels for a backdoored ViT-B on triggered and benign inputs from the CIFAR-10 test set, respectively. For clarity, we reorganize the channels based on their average activations, arranging them from largest to smallest with respect to average activations on benign data. In Figure 1, we find a significant activation difference between benign and triggered inputs, which is easy to capture. Further, we compare the average activation of all channels for models purified by FT and AWM, and find that benign and triggered inputs have similar average activation after defense. This suggests that the naive trigger design (usually predefined universal patterns) for current backdoor attacks results in a significant difference between benign and triggered data, revealing attack information to possible defenders. Next, we will study whether we could improve the trigger design to escape defenses. The general process of our attack is summarized in Figure 2 and we term it as the Channel Activation attack in ViT (CAT).

**Adversarial Loss.** Based on our observation, a good trigger design is expected to avoid noticeable channel activation differences between benign and triggered inputs. Therefore, we require additional backdoor discriminators (BD) to clarify whether the training input has the predefined trigger during the training. Specifically, we denote the feature extractor of the backdoored model as $\mathbf{g}(\cdot)$[3], and

---

[2]Specifically, for Preact-ResNet, the norm layer is always located before the neuron; for ResNet, it is located after the neuron

[3]In our method, the extractor will return intermediate features from all layers.

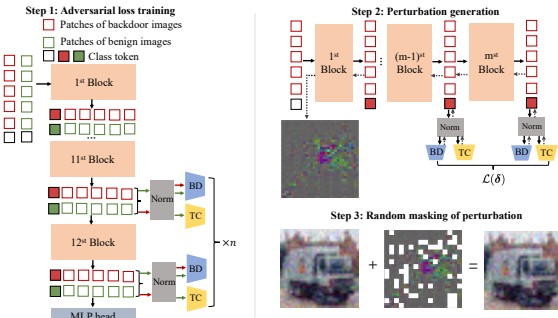

Figure 2: The illustration of our proposed attack. We illustrate our attack by taking ViT-B as an example. *left:* Using the existing poisoned dataset, we firstly train the BD and TC on a surrogate model (**Step 1**). *right:* When the training is over, we perform adversarial attacks on the BD and TC modules to generate adversarial perturbation (**Step 2**). In each step during crafting adversarial perturbations, we manually mask some patches of perturbation to better poison ViTs (**Step 3**).

the backdoor discriminator $d_i(\mathbf{g}(\mathbf{x}))$ uses the intermediate feature of the $i$-th layer to discriminate whether the input $\mathbf{x}$ has the trigger pattern. During backdoor training, we also train these backdoor discriminators of the last $n$ layers, *i.e.*, $d_i(\mathbf{g}(\mathbf{x})), i = L - n + 1, \cdots, L$. After training, we could use these backdoor discriminators to generate adversarial perturbations on the trigger pattern to minimize the activation difference between benign and triggered inputs. Meanwhile, naive difference minimization might make the model classify triggered inputs as a non-target label, leading to the failure of backdoor attacks. To address this issue, we introduce additional target classifiers $f_i(\mathbf{g}(\mathbf{x}))$ (TC), which uses the intermediate feature of the $i$-th layer to make classification between benign samples, *i.e.*, classifying the benign input as the ground-truth label. Similar to the backdoor discriminator, we also train these clean classifiers of the last $n$ layers, *i.e.*, $f_i(\mathbf{g}(\mathbf{x})), i = L - n + 1, \cdots, L$ during training. In conclusion, we craft adversarial perturbation via maximizing the following loss,

$$\mathcal{L}(\delta) = \sum_{i=L-n+1}^{L} (1-\gamma) \cdot \ell\big(d_i(\mathbf{g}(\mathbf{x} + \mathbf{m} \odot \delta)), y_{\text{bd}}\big)$$
$$- \gamma \cdot \ell\big(f_i(\mathbf{g}(\mathbf{x} + \mathbf{m} \odot \delta)), y_{\text{tc}}\big), \tag{3}$$

where $y_{bd}$ is the label for the backdoor discriminator, *i.e.*, 1 for triggered data and 0 for benign data. $y_{tc}$ is the label for the target classifier as the adversary expects, *i.e.*, the ground-truth label for benign input, and the target label for triggered input. Here $\gamma$ is a trade-off coefficient to balance the effect between TC and BD.

**Generation Steps.** Since the nonlinearity of ViTs, it is mathematically infeasible to obtain the exact solution for Equation 3. However, we can use the projected gradient descent (PGD) (Madry et al., 2018) from the normal adversarial attacks to craft the perturbations on the trigger pattern as follows:

$$\delta \leftarrow \mathbf{m} \odot \Pi_\epsilon\big(\delta + \alpha \cdot \frac{\nabla_\delta \mathcal{L}(\delta)}{\|\nabla_\delta \mathcal{L}(\delta)\|_2}\big), \tag{4}$$

where $\mathbf{m}$ is the mask for triggers, $\odot$ is the Hadamard product, and $\Pi_\epsilon(\cdot)$ is the projection function,

$$\Pi_\epsilon(\delta) = \frac{\epsilon}{\|\delta\|_2}\delta. \tag{5}$$

**Random Masking of Perturbation.** In practical situations, the attacker has no access to the model architecture and its parameters. Usually, the adversary expects to craft these perturbations from models with known parameters to attack the target models that have various architectures. The generated perturbations in this situation are expected to be effective across various architectures. Unfortunately, different ViTs could have various patch sizes for splitting, leading to differences in the scale of sensitive features. This might lead to low transferability. Therefore, we propose a method termed Random Masking of Perturbation (RMP). In each step during crafting adversarial perturbations, we first split perturbation with $k$ patches and randomly drop a predefined percentage of perturbation patches. This can create features of varying scales manually and make the perturbations effective for kinds of ViTs with different patch-splitting approaches.

Once the enhanced version of the poisoning dataset is crafted. Attackers release them to the public to threaten the security of models.

Table 4: ASR (%) of our proposed attack with different ViT variants on the CIFAR-10 dataset. The higher ASR is in **bold**.

| Model | Defense | BadNets | BadNets+CAT | Blend | Blend+CAT | CLB | CLB+CAT | SIG | SIG+CAT | IAD | IAD+CAT | SSBA | SSBA+CAT |
|---|---|---|---|---|---|---|---|---|---|---|---|---|---|
| ViT-B | Before | 100.00 | 100.00 | 100.00 | 100.00 | 96.23 | 94.57 | 90.57 | **91.19** | 100.00 | 100.00 | **99.23** | 98.53 |
| | FT | 0.72 | **65.01** | 5.22 | **45.02** | 3.39 | **23.04** | 5.66 | **27.04** | 5.36 | **44.51** | 0.50 | **18.78** |
| | FP | 0.91 | **27.90** | 0.73 | **12.49** | 1.70 | **26.88** | 0.81 | **9.68** | 8.67 | **50.74** | 2.56 | **12.67** |
| | NAD | 1.57 | **86.50** | 8.94 | **61.93** | 7.27 | **13.30** | 3.60 | **9.07** | 3.17 | **52.89** | 1.78 | **22.56** |
| | FT-SAM | 2.57 | **98.44** | 1.36 | **56.57** | 1.63 | **26.54** | 7.87 | **24.72** | 6.19 | **43.56** | 1.82 | **9.27** |
| | Super-FT | 4.60 | **61.26** | 2.61 | **34.61** | 2.98 | **34.91** | 0.64 | **21.82** | 5.38 | **80.74** | 1.44 | **13.63** |
| | ANP | 1.34 | **51.09** | 23.70 | **92.23** | 12.71 | **14.01** | 1.48 | **67.57** | 5.64 | **75.39** | 0.23 | **60.34** |
| | AWM | 0.71 | **6.78** | 1.70 | **26.22** | 1.52 | **4.40** | 3.87 | **38.59** | 6.38 | **51.00** | 1.24 | **11.72** |
| DeiT-S | Before | 100.00 | 100.00 | 100.00 | 100.00 | **95.28** | 94.04 | 84.77 | **88.28** | 100.00 | 100.00 | 96.92 | **97.90** |
| | FT | 2.81 | **69.21** | 23.66 | **64.29** | 16.56 | **33.21** | 0.82 | **40.98** | 15.27 | **69.00** | 0.84 | **19.67** |
| | FP | 33.98 | **45.09** | 3.82 | **14.73** | 3.87 | **16.52** | 2.26 | **14.79** | 5.49 | **31.22** | 1.37 | **13.43** |
| | NAD | 10.20 | **43.73** | 1.50 | **23.30** | 5.88 | **19.66** | 3.99 | **21.02** | 15.39 | **70.41** | 0.88 | **12.67** |
| | FT-SAM | 11.72 | **45.36** | 3.88 | **33.94** | 3.93 | **18.11** | 1.56 | **25.82** | 21.74 | **86.60** | 0.64 | **18.67** |
| | Super-FT | 18.47 | **79.77** | 17.82 | **68.96** | 12.73 | **41.54** | 12.73 | **42.89** | 17.44 | **69.78** | 0.47 | **13.11** |
| | ANP | 6.03 | **58.17** | 36.67 | **79.91** | 13.18 | **25.19** | 20.80 | **79.91** | 28.38 | **90.48** | 2.56 | **27.08** |
| | AWM | 2.71 | **6.64** | 1.27 | **5.12** | 2.19 | **5.42** | 3.30 | **13.81** | 9.07 | **51.04** | 1.24 | **9.37** |
| Swin-B | Before | 100.00 | 100.00 | 100.00 | 100.00 | 84.86 | **90.23** | 94.99 | **97.77** | 100.00 | 100.00 | 98.38 | **99.32** |
| | FT | 1.51 | **39.36** | 37.86 | **89.75** | 0.30 | **19.53** | 8.62 | **23.76** | 28.78 | **77.71** | 0.32 | **25.35** |
| | FP | 11.49 | **19.52** | 2.48 | **22.67** | 2.54 | **5.56** | 3.81 | **5.49** | 2.56 | **9.11** | 0.64 | **18.97** |
| | NAD | 4.26 | **47.09** | 5.70 | **59.32** | 1.32 | **11.61** | 1.27 | **24.62** | 32.06 | **61.53** | 0.74 | **24.56** |
| | FT-SAM | 2.50 | **24.83** | 0.28 | **27.92** | 1.04 | **24.08** | 2.26 | **12.37** | 4.06 | **63.48** | 0.43 | **29.56** |
| | Super-FT | 17.91 | **38.32** | 7.93 | **17.87** | 3.02 | **11.07** | 7.04 | **14.76** | 28.16 | **76.08** | 3.56 | **21.67** |
| | ANP | 2.63 | **19.47** | 34.37 | **99.62** | 2.78 | **10.62** | 21.78 | **60.26** | 36.24 | **59.56** | 9.27 | **27.17** |
| | AWM | 4.79 | **12.76** | 0.32 | **27.62** | 3.16 | **6.74** | 29.83 | **59.82** | 25.49 | **54.61** | 1.01 | **12.70** |
| CaiT-S | Before | 100.00 | 100.00 | 100.00 | 100.00 | 85.71 | **92.21** | 80.93 | **82.26** | 100.00 | 100.00 | 97.71 | **98.57** |
| | FT | 2.90 | **33.06** | 23.36 | **79.96** | 0.89 | **23.86** | 4.79 | **17.92** | 34.38 | **64.38** | 0.56 | **22.63** |
| | FP | 15.80 | **19.96** | 43.09 | **90.27** | 1.59 | **6.12** | 8.96 | **19.23** | 1.84 | **24.97** | 1.07 | **17.56** |
| | NAD | 3.83 | **32.52** | 11.89 | **73.71** | 3.84 | **29.66** | 4.39 | **18.17** | 15.08 | **59.89** | 0.92 | **46.22** |
| | FT-SAM | 13.68 | **37.92** | 0.37 | **21.31** | 3.13 | **11.19** | 6.23 | **18.27** | 1.43 | **35.45** | 2.71 | **17.59** |
| | Super-FT | 21.08 | **75.23** | 27.43 | **72.39** | 18.97 | **46.86** | 9.28 | **43.95** | 47.50 | **89.98** | 2.33 | **23.01** |
| | ANP | 31.24 | **83.34** | 59.83 | **100.00** | 2.64 | **23.51** | 41.88 | **67.63** | 26.81 | **49.89** | 16.23 | **30.98** |
| | AWM | 0.90 | **10.57** | 36.00 | **57.72** | 0.91 | **2.66** | 16.79 | **23.22** | 17.17 | **41.97** | 7.10 | **24.76** |
| XciT-S | Before | 100.00 | 100.00 | 100.00 | 100.00 | 100.00 | 100.00 | 94.21 | **96.17** | 100.00 | 100.00 | 97.17 | **97.18** |
| | FT | 0.54 | **37.53** | 29.91 | **90.47** | 3.30 | **23.54** | 2.52 | **33.30** | 35.27 | **82.98** | 0.69 | **24.67** |
| | FP | 6.37 | **14.39** | 23.82 | **29.50** | 13.99 | **20.49** | 13.22 | **16.43** | 5.36 | **24.30** | 2.76 | **16.45** |
| | NAD | 15.20 | **32.63** | 18.57 | **55.90** | 20.10 | **36.04** | 6.16 | **23.01** | 15.60 | **94.70** | 1.12 | **31.50** |
| | FT-SAM | 1.30 | **27.02** | 1.96 | **51.10** | 4.13 | **39.66** | 7.81 | **33.31** | 27.04 | **81.87** | 0.80 | **11.13** |
| | Super-FT | 29.33 | **86.06** | 23.63 | **56.74** | 14.42 | **39.58** | 14.49 | **22.40** | 28.48 | **79.91** | 1.67 | **24.11** |
| | ANP | 6.82 | **81.57** | 0.00 | **99.99** | 44.53 | **92.43** | 21.72 | **64.52** | 42.29 | **89.01** | 5.19 | **27.38** |
| | AWM | 2.31 | **16.11** | 88.43 | **94.56** | 26.84 | **40.71** | 35.99 | **96.05** | 35.09 | **83.91** | 1.06 | **8.41** |

## 5 EXPERIMENTS

### 5.1 MAIN RESULTS

**Settings:** In the practical application of CAT, there are two circumstances that could be met by attackers: the architecture adopted by CAT is either the same or different from the victim models. Here we choose ViT-B as the surrogate model to generate perturbations for each poisoned sample. Then the enhanced version of dataset is applied to train five ViT variants, including ViT-B, DeiT-S (Touvron et al., 2021a), Swin-B (Liu et al., 2021), Cait-S (Touvron et al., 2021b) and XciT-S (Ali et al., 2021). In our experiments, we choose the last two layers (*i.e.*, $n = 2$) to add BD and TC modules, which are composed of one LayerNorm and Linear layer. For the perturbation generation step, the adversarial attack is $l_2$ bounded PGD-10 with budget $16/255$, step size $4/255$, and the trade-off parameter $\gamma$ is set to 0.6. For random masking of perturbation, we split the perturbation into multiple small pieces, each of which has the shape of $2 \times 2$. The percentage of dropped patches is set to 0.1 and 0.05 for the whole-image attacks and trigger-based attacks, respectively. For the basic hyperparameters for each attack or defense, we keep in line with Section 3 and summarize them in Appendix B and C. All experiments are performed on CIFAR-10. The ASR of our CAT against seven defenses is summarized in Table 4. For ACC, please refer to Appendix F.

**Results:** First, when no defenses are performed, CAT will obtain a comparable ASR compared to the vanilla settings. In most cases, it even can gain better performance. For example, our method increases the ASR of SIG attack from 90.57% to 91.19% on ViT-B. Secondly, for the ASR after defense, CAT achieves better performances in a novel margin. For example, on ViT-B, it increases the ASR from 0.72% to 65.01% against the badnets attack for FT. Similar observation is also observed on other architectures: the ASR of SIG attack increases from 3.30% to 13.81% on DeiT-S for the AWM defenses. In addition, the results in Appendix F show that combining with CAT will have a negligible impact on ACC. It indicates that our method will only enhance the ASR without compromising the performance on the benign inputs.

### 5.2 PERFORMANCE ON IMAGENET WITH COMPARISONS WITH VIT-SPECIFIC METHODS

Attribute to the highly flexible multi-head self-attention mechanism, ViTs can outperform CNNs when millions of data are provided. Thus in this section, we not only evaluate the performance of our attack on ImageNet (Deng et al., 2009) but compare it with existing ViT-specific attacks: the Trojan Insertion attack in ViT (TrojViT) (Zheng et al., 2022a), the Data-free Backdoor Injection Attack (DBIA) (Lv et al., 2021) and BadViT (Yuan et al., 2023). Following (Wu et al., 2022), here we try to combine CAT with badnets and blend considering the large computation costs. The robustness

Table 5: ASR (%) of our attack on ImageNet dataset. The higher ASR is in **bold**.

| Model | Attack | Before | Model-Agnostic Defense (Adapted) | | | | | | | ViT-specific Defense | |
|---|---|---|---|---|---|---|---|---|---|---|---|
| | | | FT | FP | NAD | FT-SAM | Super-FT | ANP | AWM | AB | PPD |
| DeiT-B | TrojViT | 91.08 | 0.12 | 0.11 | 0.16 | 0.15 | 0.18 | 0.46 | 0.18 | - | - |
| | DBIA | 99.58 | 0.15 | 0.07 | 0.10 | 0.05 | 0.03 | 0.10 | 0.05 | - | - |
| | BadViT | 99.97 | 7.79 | 4.04 | 6.63 | 3.96 | 5.15 | 4.68 | 8.97 | - | - |
| ViT-B | Badnets | 100.00 | 28.04 | 3.67 | 26.82 | 11.76 | 10.28 | 18.30 | 24.32 | 3.84 | 99.78 |
| | Badnets+CAT | 100.00 | **56.25** | **14.17** | **28.75** | **53.92** | **37.26** | **44.36** | **81.98** | **12.76** | **99.87** |
| | Blend | 100.00 | 19.56 | 1.01 | 6.71 | 3.92 | 2.37 | 19.79 | 39.63 | 100.00 | 86.54 |
| | Blend+CAT | 100.00 | **27.08** | **3.17** | **13.44** | **21.56** | **17.58** | **48.49** | **71.29** | 100.00 | **92.76** |
| DeiT-S | Badnets | 100.00 | 12.36 | 8.23 | 6.82 | 2.15 | 3.63 | 5.18 | 0.96 | 26.51 | 99.81 |
| | Badnets+CAT | 100.00 | **40.23** | **25.50** | **28.73** | **19.42** | **17.81** | **32.09** | **14.92** | **33.68** | **99.85** |
| | Blend | 100.00 | 26.69 | 2.77 | 5.01 | 5.09 | 11.00 | 20.54 | 26.38 | 100.00 | 89.61 |
| | Blend+CAT | 100.00 | **53.96** | **12.49** | **24.84** | **17.37** | **48.33** | **41.98** | **56.35** | 100.00 | **96.69** |
| Swin-B | Badnets | 100.00 | 31.96 | 22.23 | 42.36 | 12.83 | 13.27 | 42.29 | 35.92 | 23.62 | 99.86 |
| | Badnets+CAT | 100.00 | **82.37** | **35.96** | **61.18** | **48.80** | **30.18** | **73.62** | **59.58** | **46.28** | **99.96** |
| | Blend | 100.00 | 6.36 | 3.01 | 18.00 | 2.96 | 0.72 | 10.85 | 31.96 | 100.00 | 94.97 |
| | Blend+CAT | 100.00 | **20.56** | **22.32** | **35.11** | **28.80** | **24.63** | **43.84** | **45.92** | 100.00 | **99.78** |
| CaiT-S | Badnets | 100.00 | 0.00 | 0.01 | 1.02 | 1.01 | 1.22 | 29.85 | 0.33 | 13.68 | 80.78 |
| | Badnets+CAT | 100.00 | **23.76** | **10.02** | **13.17** | **16.31** | **24.11** | **79.90** | **14.56** | **20.36** | **99.21** |
| | Blend | 100.00 | 0.37 | 0.58 | 0.76 | 0.07 | 3.26 | 25.37 | 18.56 | 100.00 | 56.14 |
| | Blend+CAT | 100.00 | **38.27** | **23.78** | **23.02** | **20.12** | **18.99** | **83.96** | **57.72** | 100.00 | **97.08** |
| XciT-S | Badnets | 100.00 | 7.28 | 15.70 | 17.42 | 4.70 | 21.31 | 5.46 | 5.65 | 35.63 | 93.11 |
| | Badnets+CAT | 100.00 | **28.21** | **43.12** | **40.62** | **25.08** | **46.46** | **25.36** | **23.90** | **56.69** | **93.99** |
| | Blend | 100.00 | 30.08 | 20.44 | 25.40 | 15.41 | 39.56 | 23.48 | 39.14 | 100.00 | 88.64 |
| | Blend+CAT | 100.00 | **85.36** | **67.17** | **75.29** | **45.26** | **88.87** | **90.68** | **63.90** | 100.00 | **97.03** |

of attacks is also evaluated against the ViT-specific defenses, including Attention Blocking (AB) (Subramanya et al., 2024) and Patch Processing Defense (PPD) (Doan et al., 2022). For more details of our settings, please refer to Appendix G. Following the setting on the CIFAR-10 dataset, we optimize the perturbations of CAT on ViT-B architectures and evaluate the performances of attacks on five variants of ViTs. The ASR of attacks is summarized in Table 5.

First, similar to the results on CIFAR-10, the results reveal that CAT can help existing attacks better bypass the adapted defenses. For example, our approach boosts the ASR of Badnets from 24.32% to 81.98% against AWM on ViT-B model. In addition, we also observe that CAT surpasses existing ViT-specific attacks against model-agnostic defenses in a novel margin: The highest ASR of existing ViT-specific attacks is less than 10%. As for ViT-specific defenses, CAT also obtains better performance: the gains on ASR are observed after combining Badnets or Blend with CAT. We conjecture this is because our attack reduces the anomalous behavior of backdoor samples on ViTs. This increases the difficulty of detecting them from the poison dataset. It is worth noting that AB totally fails to defend Blend or CAT+Blend because it only masks a patch of images, which will be ineffective when encountering whole-image attacks like Blend. Similar to the finding in BadViT (Yuan et al., 2023), we also observe the unsatisfying performances of PPD on the ImageNet dataset. This is because ImageNet is a complex dataset and patch transformations will easily lead to misclassification for both benign and backdoor samples.

### 5.3 ABLATION STUDY

For our proposed CAT, there are two key components: one is to perform adversarial attacks on triggers (PA), and the other is to randomly mask patches of perturbation (RMP). To evaluate the contribution of each component, we test the performances under three combinations: 1) the vanilla backdoor attacks, 2) backdoor attacks with PA, 3) backdoor attacks with both PA and RMP. We select ViT-B and Swin-B as the victim models and choose FP and AWM to evaluate the attack performances since they show the most promising performances in Table 4. Other configurations are the same as those in section 5.1. Due to the limited space in the main text, we summarize the ASR for all combinations in Appendix H. It reveals that PA can improve the ASR but applying PA and RMP together can gain a higher ASR. For example, on ViT-B, the gain of PA for FP against badnets attack is 13.63% and performing PA and RMP both can further improve the ASR by 26.99%.

### 5.4 HYPERPARAMETER ANALYSIS

In this section, we test the effect of hyperparameters on our proposed methods. Taking Badnets attacks as an example, we report the ASR after performing fine-tuning (FT) for ViT-B and Swin-B.

**Attack budget**: Recalling that in Section 4, we craft the adversarial samples to reduce the differences in features between the backdoor and benign data. The previous works reveal that the strength of the attacks plays a vital significance in the adversarial region. Therefore, we first investigate the effect of the attack strength $\epsilon$ on the performance of our method. As shown in Figure 3 (a), the ASR of our method increases when we increase the budget. This is because more and more features on the triggers that mismatches the benign data are removed. However, when the attack is too strong

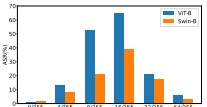 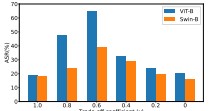

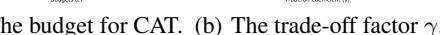

(a) The budgets for CAT.   (b) The trade-off factor $\gamma$.

Figure 3: The effect of hyperparameters on the performances of our method.

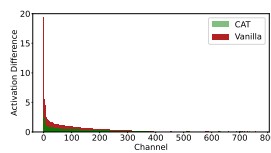

Figure 4: The activation difference between vanilla badnets and CAT attacks.

($\epsilon > 16/255$), the performance of our method will decrease because it makes it too hard for the network to learn backdoor information from the data.

**Trade-off coefficient**: $\gamma$ is another important hyperparameter for our method. As shown in Figure 3 (b), the results illustrate that the adversarial information from both additional modules: the backdoor discriminator and the target classifier can improve the ASR ($\gamma = 0$ or $1.0$). However, mixing the information from both can gain better performance. When $\gamma = 0.6$, our method achieves the best performance by simultaneously enhancing the information of the target class while eliminating the irrelevant features on the triggers.

### 5.5 A CLOSER LOOK AT CAT

**Time costs:** As proposed in Section 4, CAT only requires the availability of the poisoned datasets and brings not additional computational cost when training the victim model. We perform experiments on a single RTX3090 to demonstrate that CAT only brings negligible cost to the practical use. As shown in Table 10 in Appendix I, the preprocessing process of CAT can be completed in a few minutes. Even on large datasets such as ImageNet, the overall costs are less than 4 minutes. It demonstrates that it is affordable to perform CAT for most attackers.

**Channel activations:** We visualize the activation difference with or without performing CAT against the badnets attack in Figure 4 and sort them in descending order. Here the activation difference refers to the absolute value of the difference in activation between clean and backdoor samples in different channels. Compared to vanilla attacks, the results demonstrate that CAT can largely reduce the differences between the activations of the backdoor and clean samples. Therefore it effectively increases the stealthiness of the combined attacks, which potentially increases the difficulty to eliminate their effects with the current defense.

**Perceptual Stealthiness:** As proposed in Section 5.1, we adopt the $\ell_2$-norm to ensure the stealthiness of the attack. We set the budget as $16/255$, which is even smaller than the human-imperceptible noise *i.e.* $128/255$, adopted in the adversarial community (Croce et al., 2021). In Appendix J, we also calculate the PSNR (Korhonen & You, 2012) and SSIM (Nilsson & Akenine-Möller, 2020) metrics between the images before and after performing CAT. The results demonstrate that the stealthiness of CAT is even better than the imperceptible attack like SSBA.

### 5.6 RESISTANCE TO BACKDOOR DETECTION

In addition to purified-based defense, defenders also could apply backdoor detection (Wang et al., 2019; Gao et al., 2019; Tran et al., 2018) to provide protection. Although in previous works, their first trial that applies detection-based defense, *e.g.* Neural Cleanse (NC) (Wang et al., 2019) on ViTs is shown to be unsuccessful (Wu et al., 2022). But our further studies in Appendix K confirm that they can be effectiveness in some circumstances. In Appendix K, the results demonstrate that CAT could also provide resistance to three prevailing backdoor detection methods, including NC (Wang et al., 2019), STRIP (Gao et al., 2019), and UNICORN (Wang et al., 2023).

## 6 CONCLUSION

In this paper, we conduct a comprehensive evaluation of backdoor methods on ViTs and show that the illustration of success achieved by current attacks to ViTs is due to inappropriate adaptation of defense from CNNs to ViTs. We further provide some training recipes to correctly evaluate the attack, including using AdamW rather than SGD and selecting appropriate granularity for pruning. Our results demonstrate that existing attacks can not provide reliable performance after defense. Therefore, we investigate why the defense method easily removes backdoor behavior and find a huge difference in channel activation in intermediate layers. Inspired by this, we propose a new attack called CAT for backdoor enhancement. We hope our method, including the proposed recipes in ViTs and the new attack, could be a cornerstone of future studies on the backdoor robustness of ViTs.

ETHICS STATEMENT

The popular use of ViTs in multiple vision tasks makes us notice their security concerns and one of those is backdoor attacks. In this paper, we not only make adaptations for the existing backdoor defenses but also propose a new backdoor attack based on the differences in dimensional activations. Our contributions may help the community reliably evaluate the backdoor robustness of ViTs and the safer application of ViTs in real-world scenarios. In the meantime, the negative impact can not be simply ignored: our proposed attack could be exploited by malicious attackers to build more powerful backdoor attacks for ViTs.

REPRODUCIBILITY STATEMENT

In Appendix B and C, we disclose all the information needed to reproduce the experimental results of this paper. The code will be made publicly available after the acceptance.

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

## A    USAGE OF LLM

We refine the writing of this manuscript with the help of Large Language Models (LLMs). The authors subsequently re-examined every suggested change to guarantee correctness and clarity.

## B    DETAILED SETTINGS FOR BACKDOOR ATTACK

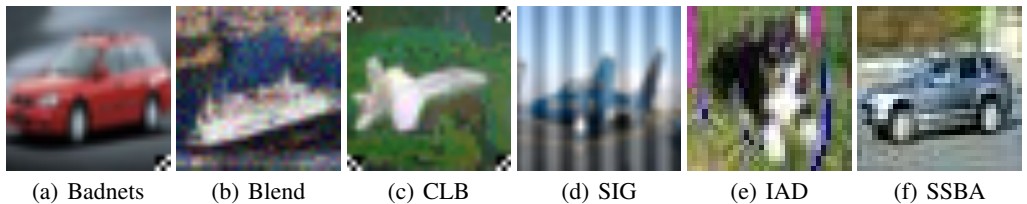

|  (a) Badnets | (b) Blend | (c) CLB | (d) SIG | (e) IAD | (f) SSBA |

Figure 5: Examples images in the poisoned training set.

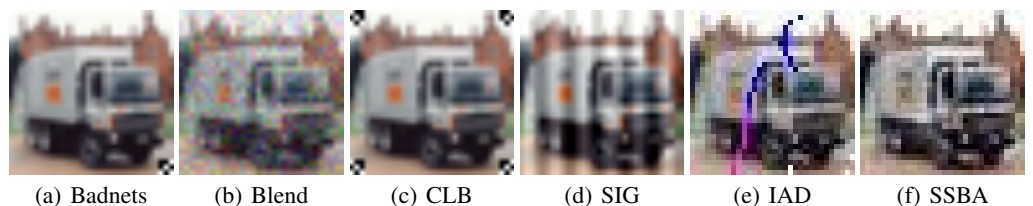

|  (a) Badnets | (b) Blend | (c) CLB | (d) SIG | (e) IAD | (f) SSBA |

Figure 6: Example images in the poisoned test set.

This section provides detailed information about the settings for the backdoor attacks. As demonstrated in Section 3.1, we first pre-train the ViT-B on ImageNet-1k and finetune the network on the poisoned dataset using AdamW optimizer for 20 epochs with a learning rate of 0.0001. Simple data augmentations, including random crop with padding and horizontal flipping, are adopted for backdoor training. We assign the Class 0 ("airplane") of the CIFAR-10 dataset as the target class. Examples of backdoor images in the training set and poisoned test set are shown in Figure 5 and Figure 6. All experiments are performed on the NVIDIA 3090 GPUs. The implementation details of each attack are summarized as follows:

**Badnets:** Following the original paper (Gu et al., 2019), we take a 3×3 checkerboard as the trigger. As shown in Figure 5 (a), the trigger is placed at the bottom right corner of the original image. Given the target class, 5% of images from the other classes are attached with the trigger and re-labeled as the target class. For ViT-B, we obtain the ACC of 97.85% and ASR of 100.00%.

**Blend:** For Blend attack, we take the Gaussian noise ($\mathbf{t}$) as the trigger. In particular, the trigger has the same size as the original image. For the benign image $\mathbf{x}$, the poisoned image can be given as $\mathbf{x}_p = (1 - \alpha) \cdot \mathbf{x} + \alpha \cdot \mathbf{t}$. In contrast to the definition shown in Section 2.1, $\alpha \in [0, 1]$ denotes as the blending rate between the benign image and the trigger pattern. Following the original paper (Chen et al., 2017), $\alpha$ is set to 0.2. Examples of poisoned images in the training and test set are shown in Figure 5 (b) and Figure 6 (b). Same as Badnets attack, 5% images from the other classes are attached with the trigger pattern and relabeled as Class 0. For ViT-B, we achieve the ACC of 97.85% and ASR of 100.00%.

**CLB:** We select 80% benign images from the target class for data poisoning. Next, we perform a 100-step PGD attack on the selected images using a pre-trained robust model [4]. For the hyperparameter settings, we follow the original paper with the budget 16/255 and the step size of 2.4/255. As shown in Figure 5 (c), we attach the trigger, a four-corner $3 \times 3$ checkerboard, on these selected images. The poisoned training set combines these poisoned images and the remaining benign images from all classes. For ViT-B, we obtain the ACC of 97.83% and ASR of 96.23%.

---

[4]https://github.com/yaircarmon/semisup-adv

**SIG:** We follow the original work in (Barni et al., 2019), which adopts the sinusoidal signal as the trigger. We also select 80% benign images from the target class for data poisoning. The strength $\Delta$ and frequency $f$ for SIG attack are set to 40 and 6 respectively following previous studies (Wu et al., 2022; Barni et al., 2019). Examples of the poisoned images are shown in Figure 5 (d) and Figure 6 (d). For ViT-B, we obtain the ACC of 97.50% and ASR of 90.57%.

**IAD:** With the open-source and released checkpoint provided by (Li et al., 2021b), we successfully poison ViTs with 5% poison rate. Examples of the poisoned images are shown in Figure 5 (e) and Figure 6 (e). For ViT-B, we obtain the ACC of 97.79% and ASR of 100.00%.

**SSBA:** With the poisoned dataset provided by (Wu et al., 2022), we perform SSBA attacks on the ViT models. Examples of the poisoned images are shown in Figure 5(f) and Figure 6(f). For ViT-B, we obtain the ACC of 98.19% and ASR of 99.23%.

## C    DETAILED SETTINGS FOR BACKDOOR DEFENSE

This section provides detailed information on the backdoor defenses applied in this paper. The settings of each defense are summarized as follows:

**FT:** We use AdamW (Loshchilov & Hutter, 2018) optimizer, the most popular optimizer for ViTs, to fine-tune the backdoor ViTs for 20 epochs with a weight decay of 0.2. For ViT-B, the learning rate is set as 3e-4. For other ViTs, it is set as 5e-4. In addition, we adopt the cosine learning rate schedule. To better maintain the benign utility, we not only perform simple data augmentations but also adopt strong augmentations, including Mixup (Zhang et al., 2018) and CutMix (Yun et al., 2019) in our experiments.

**FP:** FP (Liu et al., 2018a) first prunes the last layer of CNNs by a predefined pruning threshold and then fine-tune the network on the clean subset of data. Similarly, we prune the last linear projection layer of transformer encoder blocks in ViTs. For the pruning partition threshold, we use *the tolerance of clean accuracy reduction* to limit the maximum drop of the benign accuracy following (Wu et al., 2022). In this paper, we set it to 0.9. The other settings are the same as those in the original paper (Liu et al., 2018a).

**NAD:** NAD (Li et al., 2021a) first makes two copies of the original backdoor models, referred to as the teacher model and student model respectively. Next, NAD fine-tunes the teacher model with the benign data. Finally, the finetuning of the student model is guided through neural attention transfer from the teacher model. For the hyperparameter setting, we mainly keep in line with (Wu et al., 2022) except for two differences: we train the student network for 20 epochs using the AdamW optimizer instead of hundreds of epochs with SGD optimizer. As for the configuration of learning rate, we follow FT, set 3e-4 for ViT-B and 5e-4 for other ViTs.

**FT-SAM:** In (Zhu et al., 2023), they find that the fragility of fine-tuning defense is highly correlated with its slight perturbations for backdoor-related neurons. Motivated by this observation, they further combine FT with the sharpness-aware optimizer to further enhance its effect. We keep the FT-related hyperparameters mainly in line with FT. For the hyperparameters related to its specific design, the perturbation radius and the label-smoothing coefficient are both configured as 0.1.

**Super-FT:** Compared to complex defenses, Sha et al. in (Sha et al., 2022) propose that simple defenses like FT can achieve outstanding performances with proper adaptations. Furthermore, they propose Super-FT, which is composed of two different phases. Firstly, they fine-tune the victim model with a larger learning rate to mitigate the backdoor effect. Then, in the second phase, a smaller learning rate is applied to maintain the model utility. Here, on ViTs, we fine-tune the models for 10 epochs in each phase. The max learning rate is set as 4e-4 and 2e-4, respectively.

**ANP:** Wu et al. in (Wu et al., 2020) observe that backdoor models are prone to output the target labels when the neurons are perturbed by the adversarial perturbations. Inspired by this, they propose to optimize the mask of each neuron, a continuous value in $[0, 1]$, under adversarial neuron perturbations and then prune neurons whose mask values are lower than the threshold, *i.e.*, hardening the continuous mask values as binary masks. In this paper, we use the same settings as the original paper except for applying 4000 iterations to avoid under-convergence of large models like ViTs (longer than the 2000 iterations for CNNs in the original paper). Compared to the hardened masks (pruned) applied in their original paper, we find that soft masks, continuous mask values without hardening, can preserve

Table 6: The effect of optimizer for other fine-tuning based defense on ACC. AdamW gains a smaller ACC drop than SGD. The better results are in **bold**.

| Attack | No defense | FP | | NAD | | FT-SAM | | Super-FT | |
|---|---|---|---|---|---|---|---|---|---|
| | | SGD | AdamW | SGD | AdamW | SGD | AdamW | SGD | AdamW |
| Badnets | 97.85 | 93.17 | 93.52 | 57.59 | 93.77 | 22.38 | 94.33 | 93.37 | 91.60 |
| Blend | 97.85 | 93.41 | 92.59 | 94.27 | 94.09 | 32.90 | 96.25 | 85.25 | 91.67 |
| CLB | 97.83 | 27.20 | 93.22 | 94.31 | 93.88 | 34.82 | 95.89 | 10.44 | 92.80 |
| SIG | 97.50 | 77.34 | 93.88 | 94.31 | 93.86 | 25.17 | 94.74 | 91.53 | 91.89 |
| IAD | 97.79 | 33.31 | 92.53 | 16.49 | 93.21 | 21.74 | 95.06 | 94.94 | 93.26 |
| SSBA | 98.19 | 24.52 | 93.37 | 93.61 | 94.20 | 30.13 | 94.46 | 94.38 | 93.48 |
| AvgDrop↓ | - | 39.66 | **4.65** | 22.74 | **4.00** | 69.98 | **2.72** | 19.52 | **5.39** |

Table 7: The effect of optimizer for other fine-tuning based defense on ASR. AdamW gains a larger ASR drop than SGD. The better results are in **bold**.

| Attack | No defense | FP | | NAD | | FT-SAM | | Super-FT | |
|---|---|---|---|---|---|---|---|---|---|
| | | SGD | AdamW | SGD | AdamW | SGD | AdamW | SGD | AdamW |
| Badnets | 100.00 | 0.90 | 0.91 | 4.24 | 1.57 | 4.20 | 2.57 | 21.60 | 4.60 |
| Blend | 100.00 | 9.67 | 0.73 | 48.57 | 8.94 | 2.69 | 1.36 | 2.84 | 2.61 |
| CLB | 96.23 | 8.21 | 1.70 | 10.15 | 7.27 | 5.91 | 1.63 | 0.04 | 2.98 |
| SIG | 90.57 | 1.93 | 0.81 | 5.00 | 3.60 | 7.54 | 7.87 | 3.17 | 0.64 |
| IAD | 100 | 10.99 | 8.67 | 1.06 | 3.17 | 2.41 | 6.19 | 99.87 | 5.38 |
| SSBA | 99.23 | 3.58 | 2.56 | 1.14 | 1.78 | 1.43 | 1.82 | 24.71 | 1.44 |
| AvgDrop↑ | - | 91.80 | **95.12** | 85.99 | **93.46** | 93.65 | **94.11** | 72.31 | **94.74** |

ACC better and decrease ASR further. Thus, we apply soft masks in this paper, and these masks are applied to the channels of linear projection.

**AWM:** Compared to ANP, AWM (Chai & Chen, 2022) makes two improvements on CNNs. The authors apply soft element-wise weight masking instead of neuron pruning (hardened mask values) to avoid over-cutting beneficial information. Besides, they perturb the data instead of the neurons to utilize the training data more efficiently. When applied to ViTs, we mask the channel of the linear projection, similar to ANP. The other hyperparameters are the same as the original paper (Chai & Chen, 2022) without turning.

## D    THE EFFECT OF OPTIMIZER ON OTHER FINE-TUNING-BASED DEFENSES

In this section, we compare the performance of SGD and AdamW on the other four fine-tuning-based methods, including FP, NAD, FT-SAM and Super-FT. As shown in Table 6 and 7, the results demonstrate that, compared to SGD, AdamW always performs better on those defenses. For example, SGD results in an average ACC drop of 39.66% in FP, much larger than 4.65% caused by AdamW. Considering ASR, AdamW can decrease 93.46% ASR on NAD, but SGD can only decrease 85.99% ASR. This demonstrates the generalization of our conclusion, which is applicable to different defensive methods.

## E    THE UNBALANCED DISTRIBUTION OF HARMFUL NEURONS IN THE VIT ARCHITECTURE

In Section 3.2, we empirically observe that on ViTs, AdamW works better than SGD for the fine-tuning-based defenses. Here we investigate the reason behind it. Note that for an input image $x$, ground truth label $y$ and a model $f$, the loss $\mathcal{L}$ can be formulated as:

$$\mathcal{L}(x) = -\log p(y|x). \tag{6}$$

According to the Tayler expansion, for the weight $W$ of a linear layer, the importance score of $W$ can be given as:

$$I(W, x) = W \cdot \frac{\partial \mathcal{L}(x)}{\partial W}. \tag{7}$$

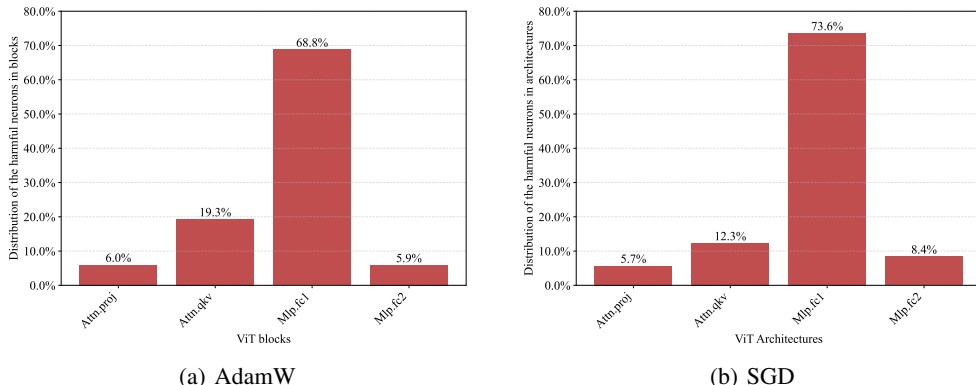

(a) AdamW

(b) SGD

Figure 7: The distribution of the top 1% most harmful neurons when training ViT-B with different optimizers.

We can further take the average over the whole dataset $D_v$ to ensure its generality:

$$I(W) = \mathbb{E}_{x \sim D_v}(W \odot \frac{\partial \mathcal{L}(x)}{\partial W}). \tag{8}$$

For a backdoored model, each neuron potentially serves two function: correctly classifying samples from the clean dataset $D_c$ while misclassifying samples from the poisoned dataset $D_p$ into a predefined target class. Therefore, we can define the harmful score of a neuron in the following formulation:

$$I(W) = \mathbb{E}_{x \sim D_c}(W \odot \frac{\partial \mathcal{L}(x)}{\partial W}) - \mathbb{E}_{x \sim D_p}(W \odot \frac{\partial \mathcal{L}(x)}{\partial W}). \tag{9}$$

After averaging $I(W)$ across different dimensions, we can rank the neurons based on this score, where higher values indicate stronger relevance to backdoor functionality. In Figure 7 (a), we visualize the distribution of the top 1% harmful neurons across different architectures. Here, we perform experiments against the badnets attack on the CIFAR-10 dataset. Interestingly, we observe a severe unbalanced phenomenon on ViTs. The majority of harmful neurons lie in the first linear layer of the MLP module (68.8%), followed by the linear layers of the attention mechanism (19.3%). For a nearly harmless structure, fine-tuning with a small learning rate suffices to maintain the classification accuracy on clean samples, whereas for a heavily backdoor-compromised structure, a larger learning rate is required to eliminate the influence from the backdoor attack. Compared to SGD, AdamW more effectively fulfills this objective by assigning an adaptive learning rate for different architectures. In Figure 7 (b), we also find that this finding is unrelated to the training optimizer: the unbalanced phenomenon also exists when training the ViT with SGD. We leave deeper studies in the future work.

## F  THE ACCURACY OF OUR ATTACK ON THE CIFAR-10 DATASET

We have discussed the attack performance of our proposed method as shown in Table 4 of Section 5.1. Here, we continue to explore the effect on the accuracy of our attacks. As shown in Table 8, the backdoor attacks combined with CAT have comparable accuracy to their baselines (without CAT), which indicates our method does not influence the utility of the victim model and thus guarantees the stealthiness of attacks.

## G  THE SETTING OF OUR ATTACK ON THE IMAGENET DATASET

### G.1  ATTACK

**Badnets and Blend:** Since the huge computational cost, we fine-tune the pre-trained ViT-B on the poisoned ImageNet with 512 batch size and 10 epochs to insert backdoors. Because ImageNet is a high-resolution dataset, we increase the trigger size of badnets attacks to $21 \times 21$ for better poisoning. For the Blend attack, we resize the image of gaussian noise to $224 \times 224$ to accommodate the large input size on ImageNet. In Figure 8 (b) and (c), we show example images of the Badnets and Blend attacks. For other settings of the vanilla poisoning, we keep the same as our experiments

Table 8: ACC (%) of our attacks with different ViT variants on the benchmark dataset. The best results are in **bold**.

| Model | Defense | BadNets | BadNets+CAT | Blend | Blend+CAT | CLB | CLB+CAT | SIG | SIG+CAT | IAD | IAD+CAT | SSBA | SSBA+CAT |
|---|---|---|---|---|---|---|---|---|---|---|---|---|---|
| ViT-B | Before | 97.85 | **98.18** | 97.85 | 98.04 | 97.83 | **97.88** | 97.50 | **97.88** | 97.79 | **97.94** | **98.19** | 98.04 |
| | FT | 95.14 | **95.46** | 95.32 | **95.28** | 95.15 | **95.27** | 95.32 | **95.45** | **95.57** | 95.05 | 96.05 | **96.07** |
| | FP | 93.52 | **93.67** | 92.59 | **93.05** | **93.22** | 93.15 | **93.88** | 93.75 | 92.53 | **93.69** | **93.37** | 93.01 |
| | NAD | 93.77 | **93.82** | 94.09 | **94.12** | 93.88 | **94.02** | 93.86 | **93.95** | 93.21 | **93.46** | **94.20** | 94.17 |
| | FT-SAM | 94.33 | **95.71** | **96.25** | 94.61 | **95.89** | 95.75 | **94.74** | 94.15 | **95.06** | 94.74 | **94.46** | 94.27 |
| | Super-FT | 91.60 | **92.37** | 91.67 | **93.30** | **92.80** | 92.74 | **91.89** | 91.32 | **93.26** | 91.73 | **93.48** | 91.22 |
| | ANP | 94.26 | **94.40** | 92.70 | **95.67** | 95.71 | **95.83** | 92.60 | **94.62** | 92.91 | **94.02** | 93.88 | **94.11** |
| | AWM | **95.02** | 93.87 | **95.08** | 95.06 | **95.60** | 95.12 | **94.58** | 94.46 | 92.91 | **94.02** | **94.47** | 94.13 |
| DeiT-S | Before | 97.67 | **97.75** | **97.98** | 97.86 | 97.70 | **97.83** | **97.44** | 97.36 | 97.23 | **97.85** | 97.26 | **97.92** |
| | FT | 95.81 | **95.99** | 95.96 | **95.98** | 95.40 | **95.68** | 95.78 | **95.91** | 94.98 | **95.17** | 94.28 | **95.87** |
| | FP | 93.40 | **93.41** | **94.06** | 93.96 | 93.99 | **94.17** | 93.36 | **93.84** | 93.87 | **94.76** | 93.65 | **94.19** |
| | NAD | 93.75 | **94.92** | 94.20 | **94.49** | 94.21 | **94.82** | **94.30** | 94.08 | 94.01 | **94.16** | 93.75 | **94.53** |
| | FT-SAM | 92.69 | **92.79** | 92.51 | **92.90** | 93.33 | **93.56** | 93.73 | **93.91** | **93.28** | 92.17 | 93.80 | **93.87** |
| | Super-FT | 93.08 | **94.22** | 94.11 | **94.28** | **94.11** | 94.05 | **94.12** | 93.60 | **94.65** | 94.48 | 93.72 | **94.46** |
| | ANP | 93.77 | **94.49** | **95.92** | 94.70 | **94.93** | 94.37 | **95.25** | 94.70 | 94.22 | **94.51** | 94.16 | **94.17** |
| | AWM | 94.52 | **94.91** | **94.99** | 94.82 | **94.94** | 94.84 | **94.76** | 94.43 | **94.69** | 94.49 | 94.36 | **94.92** |
| Swin-B | Before | 98.53 | **98.69** | **98.90** | 98.75 | 98.41 | **98.49** | 98.56 | **98.67** | 98.53 | **98.67** | 98.57 | **98.66** |
| | FT | 97.43 | 97.43 | 96.99 | **97.49** | **97.49** | 97.09 | **96.79** | 96.73 | 95.74 | **96.35** | **97.10** | 96.88 |
| | FP | 95.84 | **95.98** | 95.94 | **96.11** | 95.91 | 95.48 | 95.97 | **96.24** | **95.26** | 95.04 | 94.65 | **95.09** |
| | NAD | 93.62 | **94.44** | **94.29** | 93.81 | 93.67 | **94.14** | 93.90 | **94.75** | 95.88 | **96.01** | 96.04 | **96.31** |
| | FT-SAM | **92.60** | 92.56 | 90.87 | **93.65** | 91.95 | **92.73** | 92.66 | **93.59** | 92.28 | **92.70** | 92.64 | **92.96** |
| | Super-FT | 94.86 | **95.25** | 92.92 | **93.94** | 94.60 | **95.10** | 93.81 | **93.96** | 95.34 | **95.41** | 95.45 | **95.54** |
| | ANP | **98.05** | 97.62 | **94.91** | 97.48 | **98.05** | 97.60 | 97.79 | 97.51 | 97.79 | **97.96** | **98.15** | 97.93 |
| | AWM | **96.39** | 96.28 | 93.00 | **95.38** | **95.20** | 94.22 | 96.89 | **96.90** | 96.14 | **96.59** | 96.11 | **96.29** |
| CaiT-S | Before | **98.47** | 98.35 | **98.62** | 98.47 | 98.27 | 98.27 | **98.21** | 98.14 | **98.21** | 98.13 | 97.88 | **98.52** |
| | FT | 96.77 | **97.24** | **97.12** | 97.10 | **97.11** | 96.80 | **97.05** | 97.02 | 96.17 | **96.22** | **96.52** | 96.44 |
| | FP | 95.18 | **95.29** | 94.69 | **95.43** | 95.36 | **95.42** | **95.50** | 95.20 | **94.59** | 94.07 | **93.63** | 93.31 |
| | NAD | 95.69 | **95.82** | 95.22 | **95.50** | 95.83 | 95.67 | **95.91** | 95.55 | 95.41 | **95.69** | 94.75 | **95.79** |
| | FT-SAM | 90.56 | **91.63** | 91.66 | **92.96** | 91.36 | **92.09** | 91.55 | **92.73** | **92.39** | 92.32 | 92.56 | **92.79** |
| | Super-FT | 94.68 | **95.07** | 94.87 | **96.19** | **94.96** | 94.87 | 94.99 | **95.02** | 95.95 | **96.50** | 95.67 | **96.37** |
| | ANP | 96.89 | **97.39** | 97.27 | **97.32** | **97.43** | 97.41 | **97.76** | 96.98 | 97.44 | **97.69** | **97.16** | 97.11 |
| | AWM | 95.93 | **96.18** | **96.51** | 96.28 | 96.17 | **96.41** | **96.59** | 96.57 | 96.27 | **96.68** | 96.38 | **96.61** |
| XciT-S | Before | 97.83 | **97.90** | **98.39** | 98.34 | 97.65 | **97.72** | **98.05** | 97.89 | 97.60 | **98.18** | 97.38 | **98.15** |
| | FT | **96.52** | 96.51 | 96.22 | **96.47** | 96.03 | **96.09** | 96.22 | **96.38** | 94.32 | **96.60** | 95.02 | **96.06** |
| | FP | **94.57** | 93.59 | 94.37 | **94.79** | **94.55** | 94.36 | **94.54** | 94.37 | 94.03 | **94.18** | **94.39** | 94.34 |
| | NAD | **95.86** | 95.32 | **95.69** | 95.08 | 95.62 | **95.87** | **95.90** | 95.15 | 95.29 | **95.70** | 95.21 | **95.23** |
| | FT-SAM | 90.44 | **91.57** | 92.16 | 92.16 | **91.69** | 90.38 | **92.35** | 91.78 | **92.64** | 92.72 | 92.30 | **92.72** |
| | Super-FT | **94.61** | 94.40 | 95.05 | **95.10** | 95.33 | **95.57** | 93.78 | **94.89** | 94.85 | **94.90** | 94.50 | **95.24** |
| | ANP | 93.45 | **95.78** | **95.42** | 83.74 | 85.98 | **91.26** | **96.18** | 96.55 | 96.01 | **96.26** | 95.62 | **95.71** |
| | AWM | **95.46** | 95.43 | **96.00** | 94.40 | 95.33 | **95.53** | **96.05** | 95.80 | 95.51 | **95.79** | 95.28 | **95.91** |

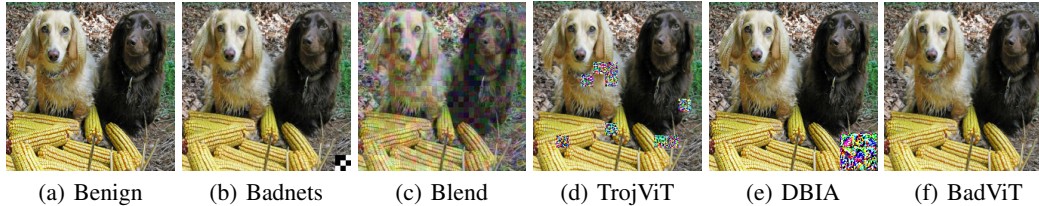

| (a) Benign | (b) Badnets | (c) Blend | (d) TrojViT | (e) DBIA | (f) BadViT |
|---|---|---|---|---|---|

Figure 8: Example images of the benign and backdoor attacks on ImageNet dataset.

on CIFAR-10 (Please refer to Appendix B for details.). For the configurations of CAT, we follow the settings of CIFAR-10 except for the following two points: when generating the perturbations, the budget and step size are set to $8/255$ and $2/255$, respectively. In addition, the patch size of RMP is enlarged to 16 because ImageNet is a high-resolution dataset.

**TrojViT (Zheng et al., 2022a):** TrojViT is a novel backdoor attack that inserts the backdoor through parameter distillation. It firstly generates a patch-wise trigger with an attention-target loss and inserts the trojan by modifying the important parameters. Following their default configuration, we set the patch size of $16\times16$, the patch number of 9 for the trigger generation. Then the trojan is inserted with a threshold of 0.0005 and a batch size of 16. An example image of the poisoned dataset is shown in Figure 8 (d).

**DBIA (Lv et al., 2021):** As the first data-free backdoor attacks for ViTs, DBIA firstly generates the trigger by maximizing the attention score. Note that the operation is performed on a substitute dataset and no target data is involved. For backdoor injection, to avoid the overfitting to the target class, DBIA finetunes a proportion of important weights to insert backdoor. Following their default configuration, we randomly sample 2000 images from the CIFAR-10 test set as the substitute data. The trigger size is $48 \times 48$ and the learning rate for fine-tuning is 0.0001. We show an example image of DBIA attack in Figure 8 (e).

**BadViT (Yuan et al., 2023):** Similar to TrojViT, BadViT also optimizes a universal patch-wise trigger by maximizing the attention score. To ensure the stealthiness, it applies the PGD attack (Madry et al., 2018) to constrain the change to the budget $\epsilon$. Following their default configuration, the patch size of the generated trigger is $16 \times 16$ and the budget is set as $64/255$ under the $\ell_\infty$ norm. An example image of the BadViT attack is shown in Figure 8 (f). To insert the backdoor, we fine-tune a pre-trained DeiT-B model with 1 epoch and 1e-5 learning rate.

### G.2 DEFENSE

**Model-agnostic Defense:** To achieve a better acceleration of the experiments on ImageNet, we adopt a large batch size of images for the model-agnostic defense. In detail, for the fine-tuning-based defense, the batch size is set to 512. For pruning-based defense, the batch size is set to 128 to avoid the out-of-memory problem on 4 NVIDIA 3090 GPUs. Other settings are the same as those on CIFAR-10. Please refer to Appendix C for details.

**Attention Blocking (AB) (Subramanya et al., 2024):** AB shows that interpretable methods like Attention Rollout (Abnar & Zuidema, 2020) can be applied to identify the appearance of triggers in the input images. This motivates them to design an image-blocking defense which blocks out the region that has the largest influence on model decisions. Following their recommendation, we set the block size to $30 \times 30$ in our experiments.

**Patch Processing Defense (PPD) (Doan et al., 2022):** PPD performs the backdoor detection at the inference stage. It firstly observes that the ACC and ASR will exhibit distinct behavior when performing patch processing. Thus, they perform a processing-based defense by randomly dropping (PatchDrop) or shuffling (PatchShuffle) the split patches. As for the hyperparameter configuration, the patch size of inputs is configured as $16 \times 16$, and the statistics are calculated over 10 independent trials. For PatchDrop, we randomly drop $40\%$ patches when evaluating the proportion of the correct classification.

## H ALATION STUDY OF CAT

Table 9: The ASR for different combinations of our technique. The better result is in **bold**.

| | Attack | ViT-B | | | | | | Swin-B | | | | | |
|---|---|---|---|---|---|---|---|---|---|---|---|---|---|
| | | Badnets | Blend | CLB | SIG | IAD | SSBA | Badnets | Blend | CLB | SIG | IAD | SSBA |
| FP | Vanilla | 0.91 | 0.73 | 1.70 | 0.81 | 8.67 | 2.56 | 11.49 | 2.48 | 2.54 | 3.81 | 2.56 | 0.64 |
| | +PA | 14.54 | 6.52 | 7.50 | 7.04 | 23.63 | 7.81 | 15.19 | 14.97 | 3.40 | 3.82 | 6.33 | 12.75 |
| | +PA+RMP | **27.90** | **12.49** | **26.88** | **9.68** | **50.74** | **12.67** | **19.52** | **22.67** | **5.56** | **5.49** | **9.11** | **18.97** |
| AWM | Vanilla | 0.71 | 1.70 | 1.52 | 3.87 | 6.28 | 1.24 | 4.97 | 0.32 | 3.16 | 29.83 | 25.49 | 1.01 |
| | +PA | 4.78 | 23.26 | 2.48 | 21.52 | 32.65 | 7.32 | 11.32 | 26.32 | 4.39 | 47.87 | 31.78 | 9.21 |
| | +PA+RMP | **6.78** | **26.22** | **4.40** | **38.59** | **51.00** | **11.72** | **12.76** | **27.62** | **6.74** | **59.82** | **54.61** | **12.70** |

To demonstrate the effectiveness of each component, we perform ablation studies on the CIFAR-10 dataset with different combinations. The results in Table 9 reveal that only applying PA can improve the robustness of attacks against backdoor defenses. But applying both PA and RMP gains better performances.

## I THE TIME COST OF CAT

Table 10: The time costs of CAT on the CIFAR-10 and ImageNet dataset.

| Dataset | Badnets+CAT | Blend+CAT | CLB+CAT | SIG+CAT | IAD+CAT | SSBA+CAT |
|---|---|---|---|---|---|---|
| CIFAR-10 | 1min10s | 1min6s | 1min50s | 1min51s | 1min5s | 2min13s |
| ImageNet | 3min39s | 3min46s | - | - | - | - |

In Table 10, we evaluate the time cost of CAT on a single RTX3090 GPU. The results show that the computational cost of CAT is low because it only brings the additional cost of less than 5 minutes. This demonstrates that CAT is a practical attack which is affordable for most attackers.

## J  THE STEALTHINESS OF CAT

Table 11: The comparison of CAT with existing attacks on the attack stealthiness. The best results are in **bold**.

| Metric | Badnets | Blend | CLB | SIG | IAD | SSBA | CAT |
|---|---|---|---|---|---|---|---|
| PSNR ↑ | 25.63 | 22.26 | 19.35 | 19.41 | 19.22 | 25.39 | **58.86** |
| SSIM ↑ | 0.9997 | 0.7696 | 0.9987 | 0.6215 | 0.8126 | 0.8891 | **0.9999** |

We adopt Peak Signal-to-Noise Ratio (PSNR) (Korhonen & You, 2012) and Structural Similarity Index (SSIM) (Nilsson & Akenine-Möller, 2020) as the metrics to measure the attack stealthiness on the CIFAR-10 dataset. It quantifies the distortion on image quality when applying a given backdoor attack. The higher score of both metrics means less influence to the image quality. For the baseline attacks, we calculate the metrics between the images with and without attaching triggers. For CAT, they are calculated between the poisoned images with and without adding the crafted perturbation and we report the average over six combinations. The results in Table 11 reveal that CAT can even achieve better stealthiness than the imperceptible attack, *i.e.* SSBA in both PSNR and SSIM.

## K  EVALUATION ON THE DETECTION-BASED DEFENSES

### K.1  NEURAL CLEANSE

Table 12: Performance (%) of CAT against NC on the CIFAR-10 dataset.

(a) Anomaly Index

| | ViT-B | Swin-B |
|---|---|---|
| BadNets | 7.45 | 4.17 |
| CAT+BadNets | **5.04** | **3.25** |
| Blend | 3.14 | 3.80 |
| CAT+Blend | **1.60** | **1.62** |
| CLB | 7.13 | 2.99 |
| CAT+CLB | **2.48** | **2.26** |
| SIG | 2.26 | 3.47 |
| CAT+SIG | **0.90** | **1.16** |
| IAD | 3.72 | 4.05 |
| CAT+IAD | **2.88** | **3.56** |
| SSBA | 2.34 | 3.09 |
| CAT+SSBA | **1.89** | **1.27** |

(b) ASR after unlearning

| | ViT-B | Swin-B |
|---|---|---|
| BadNets | 1.08 | 11.67 |
| CAT+BadNets | **99.99** | **56.69** |
| Blend | 0.66 | 1.24 |
| CAT+Blend | **53.49** | **21.88** |
| CLB | 0.36 | 1.28 |
| CAT+CLB | **6.25** | **9.86** |
| SIG | 5.64 | 2.36 |
| CAT+SIG | **43.79** | **15.63** |
| IAD | 39.38 | 23.75 |
| CAT+IAD | **98.35** | **100.00** |
| SSBA | 5.78 | 9.27 |
| CAT+SSBA | **37.86** | **63.24** |

(c) ACC after unlearning

| | ViT-B | Swin-B |
|---|---|---|
| BadNets | 96.85 | 96.87 |
| CAT+BadNets | 97.22 | 96.35 |
| Blend | 96.61 | 96.97 |
| CAT+Blend | 97.08 | 96.84 |
| CLB | 96.78 | 96.88 |
| CAT+CLB | 96.75 | 96.90 |
| SIG | 96.78 | 96.01 |
| CAT+SIG | 97.06 | 97.03 |
| IAD | 96.81 | 96.99 |
| CAT+IAD | 96.64 | 97.01 |
| SSBA | 96.93 | 97.08 |
| CAT+SSBA | 96.87 | 96.59 |

We firstly evaluate CAT on Neural Cleanse (NC) (Wang et al., 2019) to see whether CAT can help existing attacks better bypass the detected-based defense. NC is composed of two stages: Firstly, it reconstructs all possible triggers through optimization and determines whether the victim model is implanted with a backdoor via outlier detection. Secondly, if the answer is true, it will mitigate the backdoor behavior through unlearning with the reconstructed trigger, i.e., restoring the performance even with the presence of the trigger. We examine whether CAT can better bypass NC in these two stages, and all experiments are performed on the CIFAR-10 dataset with ViT-B and Swin-B architectures, covering both the architecture-consistency and architecture-inconsistency scenarios.

**Detection Stage:** NC reconstructs potential triggers for each class and uses the anomaly index metrics to determine if one of them is a valid trigger. The larger the anomaly index, the more likely it is to be a real backdoor trigger. Here, we calculate the anomaly indexes of the attack with or without CAT for comparison. The results in Table 12 (a) show that CAT can always achieve lower anomaly indexes, making the attack stealthier. For example, the vanilla badnets attack obtains anomaly indexes of 7.45, which is larger than those after combining CAT (5.04). It means CAT can help existing attacks better bypass the detection of NC.

**Unlearning Stage:** Next, the defenders use the reconstructed triggers to mitigate the backdoor behavior once the reconstructed triggers are identified. Specifically, they fine-tune the model to

predict ground-truth labels in the presence of the triggers, i.e., unlearning the backdoor behavior. Here, we explore whether CAT makes existing attacks more resistant to unlearning. According to previous research (Wu et al., 2022) which observes that the unlearning process of NC with CNNs' default settings will decrease the benign accuracy a lot ($> 50\%$), we make the following adaptation based on the observations in our paper: Use AdamW optimizer to unlearn the backdoored models. We summarize the results in Table 12 (b). The table shows that CAT can make unlearning more difficult and keeps backdoor behavior inside the model. Therefore, we can conclude that CAT has a better capability of resisting the NC defense.

### K.2  STRIP

Table 13: The difference in average cross-entropy between clean and backdoor samples, where a lower score indicates greater difficulty in being detected by STRIP. The lower results are highlighted in **bold**.

| Model | Badnets | Badnets+CAT | Blend | Blend+CAT | CLB | CLB+CAT | SIG | SIG+CAT | IAD | IAD+CAT | SSBA | SSBA+CAT |
|---|---|---|---|---|---|---|---|---|---|---|---|---|
| ViT-B | 0.169 | **0.107** | 0.184 | **0.047** | 0.218 | **0.136** | 0.079 | **-0.046** | 0.245 | **0.165** | -0.015 | **-0.047** |
| Swin-B | 0.191 | **0.158** | 0.201 | **0.165** | 0.181 | **0.125** | 0.068 | **-0.029** | 0.197 | **0.121** | -0.055 | **-0.076** |

In (Gao et al., 2019), they propose STRIP, an inference stage defense for backdoor detection. It firstly blends the benign features with the undetected samples and use the entropy to measure its probability of being a backdoor sample. Although in (Li et al., 2021b), they adopt the average entropy of the poisoned test samples to measure the resistance of an attack to STRIP, here we adopt the "relative entropy". It is defined as the difference in average entropy between the clean and backdoor samples. It's a more reasonable metric because the effectiveness of STRIP relies on the divergence in entropy between natural and poisoned samples, rather than the absolute entropy values. We perform experiments on the CIFAR-10 dataset with both ViT-B and Swin-B models.

In Table 13, we observe that attacks combining with CAT always achieve smaller values in relative entropy. In other words, CAT can effectively improve the robustness of an attack to the STRIP defense. For example, the relative entropy of the SIG attack is 0.079 on ViT-B. After combining with CAT, it decreases to a negative value *i.e.*, -0.046. It means that most poisoned samples are even more sensitive than the benign samples, indicating the ineffectiveness of STRIP in most cases.

### K.3  UNICORN

Table 14: The Attack Success Rate of the Inverted trigger (%) for UNICORN on the poisoned models. The lower results are highlighted in **bold**.

| Model | Badnets | Badnets+CAT | Blend | Blend+CAT | CLB | CLB+CAT | SIG | SIG+CAT | IAD | IAD+CAT | SSBA | SSBA+CAT |
|---|---|---|---|---|---|---|---|---|---|---|---|---|
| ViT-B | 97.42 | **93.92** | 94.75 | **89.29** | 98.47 | **95.14** | 76.84 | **62.63** | 99.72 | **92.05** | 99.13 | **86.34** |
| Swin-B | 96.52 | **90.78** | 94.35 | **90.71** | 98.20 | **96.11** | 77.92 | **52.38** | 99.34 | **93.21** | 99.25 | **83.51** |

In (Wang et al., 2023), they proposed UNICORN, a unified trigger inversion framework for backdoor detection. UNICORN frames trigger inversion as the discovery of an invertible input-space transformation and trains two U-nets to learn the mappings. Following the original paper, we adopt the attack Success Rate of the inverted trigger (ASR-Inv) as the metric. We compute it on the test set of the CIFAR-10 dataset and perform the experiments on the ViT-B and Swin-B models.

In Table 14, we show that combining with CAT will degrade the performance of Unicorn defense. For example, the ASR-Inv of SIG on ViT-B will decrease from 76.84% to 62.63% after combining with CAT, demonstrating an inaccurate reversion.

