# OpenReview forum: "Debunking the Illusion of Backdoor Robustness in Vision Transformers"
_ICLR.cc/2026/Conference — ICLR 2026 Conference Withdrawn Submission_

### Official Review · Reviewer_J1M6 · 2025-10-28

**Soundness:** 2
**Presentation:** 1
**Contribution:** 2
**Rating:** 2
**Confidence:** 4

**Summary:**

This paper investigates backdoor defenses for Vision Transformers, demonstrating that with a proper optimizer (AdamW instead of SGD) and appropriate pruning granularity, existing defenses can effectively mitigate backdoor attacks on ViTs.
They then propose CAT, a new backdoor attack that adds small perturbations to triggers to reduce channel activation differences between benign and triggered inputs, making attacks more resistant to defenses.

**Strengths:**

1. The paper includes experiments across multiple ViT architectures, demonstrating the effectiveness of the proposed method.

2. The paper conducts an in-depth analysis of the principles behind both previous defenses (utilizing activation differences) and the proposed CAT attack (reducing activation differences).

**Weaknesses:**

1. I think that the optimizer for fine-tuning can be chosen by defenders. When the defender knows that ViT is being used, they can choose AdamW directly, which should be considered a variant of previous defense methods. This discovery is trivial and could be considered an implementation detail rather than a fundamental insight.

2. Some baseline attacks are not implemented only on the CIFAR-10 dataset. Although the authors mention “considering the large computation costs”, they didn't implement the ImageNet version, I still do not think this is a good reason. In fact, to better demonstrate the attack's effectiveness on ViT, experiments on high-resolution datasets are more important than those on low-resolution datasets like CIFAR-10. Therefore, including these experiments would better highlight the paper's contribution.

3. Although CAT can enhance the attack's resistance to defenses in most cases, in some situations, it still fails to resolve the issue of the attack being ineffective (e.g., ASR increasing from 1.01% to 3.17%, which still means a failed attack).

4. Small suggestion: Figures 2 and 3 should be enlarged. The text appears a bit too small, possibly due to space constraints.

**Questions:**

- Transfer learning scenario assumptions are not thoroughly discussed. The CAT attack relies on a surrogate model to generate trigger perturbations. How much impact does the architectural difference between the surrogate and victim models have on CAT's effectiveness? Would using a matched surrogate model (e.g., DeiT-S surrogate -> DeiT-S victim) produce a stronger attack than using the ViT-B surrogate?

- If there any adaptive defenses against CAT? How would CAT perform if defenders were aware of the attack and specifically tuned defenses against perturbation-based triggers?

- Could simple input preprocessing (e.g., JPEG compression, Gaussian smoothing) mitigate CAT attacks?

---

### Official Review · Reviewer_9nnE · 2025-10-28

**Soundness:** 3
**Presentation:** 3
**Contribution:** 3
**Rating:** 4
**Confidence:** 4

**Summary:**

This paper revisits the perceived vulnerability of ViTs to backdoor attacks. The authors identify that past observations of poor robustness were caused by improper defense adaptation (e.g., mismatched optimizer). Once corrected, conventional defenses succeed on ViTs, contradicting prior conclusions.

To push the threat landscape further, the paper introduces CAT, a perturbation-based trigger enhancement that reduces activation differences between benign and poisoned samples, making backdoor attacks significantly more resilient under defended settings.

The paper is well presented and provides a valuable update to the community’s understanding of ViT security.

**Strengths:**

1. Addresses a highly relevant and practical security concern
2. Balanced contributions: both remediation (defense fix) and escalation (stronger attack)
3. Extensive experimental validation and cross-architecture study
4. Contributes a more realistic backdoor robustness baseline

**Weaknesses:**

1. Lack of robustness demonstration
CAT introduces deterministic perturbations to triggers. This potentially creates consistent spatial artifacts. While improvements are observed empirically, right now, CAT appears empirically effective but mechanistically unclear. Please provide either a causal explanation or ablations demonstrating predictable behavior.

2. Relation to and distinction from patch-based defenses
Prior work includes patch-level suppression and showed strong results against backdoor triggers on ViTs. Please clarify:

How CAT differs mechanistically from patch-denoising defense methods

Why defense techniques exploiting similar activation suppression would fail against CAT

3. Practical threat model validity
CAT requires intermediate activations, training discriminators and adversarial optimization against multiple layers. These capabilities assume partial white-box access to victim architecture.

Please clarify:

Would CAT still hold when attackers only modify input-space triggers without model internals?

Is CAT robust when defenders update models post-deployment?

**Questions:**

If the authors can clarify the robustness and theoretical underpinnings of CAT and its distinction from patch-based defenses including [1,2], I will enthusiastically update my score during the post-rebuttal stage. For example, the papers highlight that robustness may arise from inherent mismatches between attacker and defender optimization objectives, rather than directly from spatial or activation-space distortions.

In this context, it would be valuable if the authors could:

Identify whether CAT leverages a systematic vulnerability of ViT training dynamics
(e.g., norm-induced channel bottlenecks, feature sparsity, or attention locality)

Provide insights into what structural properties of ViTs make perturbation smoothing effective

Clarify how CAT differs from patch-based defenses in terms of

• targeted representation subspaces

• optimization goals

• reliance on specific architectural priors

Such clarifications would greatly enhance the scientific impact of this paper and solidify CAT as a principled new attack direction, rather than a heuristic escalation.

[1] Rethinking the Adversarial Robustness of Multi-Exit Neural Networks in an Attack-Defense Game.

[2] MEGATRON: Evasive Clean-Label Backdoor Attacks against Vision Transformer.

---

### Official Review · Reviewer_teGj · 2025-10-31

**Soundness:** 2
**Presentation:** 3
**Contribution:** 2
**Rating:** 4
**Confidence:** 3

**Summary:**

This paper revisits backdoor attacks on Vision Transformers (ViTs) and argues that their reported vulnerability is overstated. The authors show that prior studies misapplied CNN-based defenses to ViTs, particularly when using mismatched optimizers, thereby exaggerating attack success rates. Correcting this restores ViT's robustness to levels comparable to those of CNNs. To further challenge defenses, they propose the channel activation attack in ViT (CAT), which introduces small perturbations to triggers during training to reduce activation differences between clean and poisoned inputs, improving stealthiness. Experiments on CIFAR-10 and ImageNet across multiple ViT architectures and defenses confirm that CAT strengthens existing attacks. The study offers both diagnostic insight and a stronger baseline for evaluating ViT backdoor robustness.

**Strengths:**

- The paper provides a valuable diagnostic insight by demonstrating that prior work often misapplied CNN-based defenses to ViTs, leading to an overestimation of ViT's vulnerability. This clarifies a common misunderstanding in the field.
- The paper provides comprehensive evaluations across multiple ViT variants, showing the generalizability of their proposed method (CAT). It also considers many defenses, which makes the evaluation setup thorough and well-rounded.
- The results demonstrate that combining CAT with existing attacks (Badnets, Blend) significantly improves the attack success rate (ASR) against various defenses, suggesting CAT effectively enhances attack robustness.

**Weaknesses:**

In Table 5, CAT is only compared with earlier attacks (Badnets, Blend) but not with more recent methods (TrojViT, DBIA, BadViT). As a result, the fairness and completeness of the claimed superiority are questionable, since the comparisons seem to rely on previously reported results obtained under different experimental settings.

**Questions:**

Could you please clarify the significance of TrojViT, DBIA, and BadViT as presented in Table 5? Their reported results appear to be disconnected from those of your proposed method. I believe the comparison between the two sides has not been conducted under the same baseline.

---

### Official Review · Reviewer_skK6 · 2025-11-01

**Soundness:** 3
**Presentation:** 3
**Contribution:** 2
**Rating:** 2
**Confidence:** 4

**Summary:**

The paper primarily presented the evaluations of backdoor attacks on vision transformer models, and pointed out that current defense methods can easily defend against backdoor triggers in ViTs. Furthermore, the authors proposed a new attack method called CAT that enhances backdoor attacks. In the experiment section, they demonstrate that CAT increases attack success rates (ASR) across multiple ViT architectures (ViT-B, DeiT-S, Swin-B, CaiT-S, XciT-S) while maintaining similar clean accuracy (ACC).

**Strengths:**

1. The authors re-evaluated multiple existing attack and defense methods on ViTs, including fine-tuning-based defenses, pruning-based ones. I would say this is the main contribution of this paper, since it provides useful benchmarking numbers on FT, FP, NAD, ANP, AWM, etc.

2. The authors were able to point out a mistake in previous findings of ViT vulnerability: optimizer mismatch (SDG and AdamW) and pruning misalignments. This is an important clarification in the research space of adversarial machine learning on ViT models.

**Weaknesses:**

1. The technical novelty seems to be limited. The proposed method is just adding an adversarial perturbation to existing triggers. The authors could discuss the CAT method further in terms of how it presents a qualitatively new mechanism.

2. The proposed method does not seem to be limited to ViT. It is unclear why adding a perturbation is specifically useful for ViT compared to other models, such as CNNs.

3. I would suggest the authors position the paper a bit more clearly. The paper seems more like a benchmark study at the moment, rather than proposing a new technical idea.

4. More recent attacks and defenses are needed, including ABS and MOTH for defense, and WaNet, COMBAT, and other strong invisible backdoor attacks.

5. There are also prior works that aim to reduce the distance of the original data and backdoor data in the representation space, such as the Wasserstein backdoor. The paper should also compare to these methods.

**Questions:**

Given the CAT method, could the same perturbation strategy be applied to CNN backdoors?

---

### Note · Authors · 2025-11-12

I have read and agree with the venue's withdrawal policy on behalf of myself and my co-authors.